# Effects of Pelleted and Extruded Feed on Growth Performance, Intestinal Histology and Microbiota of Juvenile Red Swamp Crayfish (*Procambarus clarkii*)

**DOI:** 10.3390/ani12172252

**Published:** 2022-08-31

**Authors:** Jinjuan Wan, Qinkai Xi, Jianqing Tang, Tianji Liu, Cong Liu, Hongqin Li, Xizhang Gu, Meifang Shen, Meiqin Zhang, Jinguang Fang, Xianglong Meng

**Affiliations:** 1Freshwater Fisheries Research Institute of Jiangsu Province, Nanjing 210017, China; 2New Hope Liuhe Co., Ltd., Chengdu 610063, China

**Keywords:** xiao-long-xia, aquaculture, digestive enzyme, resident bacteria

## Abstract

**Simple Summary:**

Feed processing techniques play a vital role in aquaculture since it is important to apply the appropriate processing technique to maximize production efficiency at the lowest possible cost. In this study, two diets (pelleted and extruded feed) were formulated to fed juvenile crayfish for eight weeks. Results revealed better growth performance by feeding the extruded feed. Furthermore, feeding extruded feed also enhanced the richness of gut microbiota and reduced the proportion of harmful microbial taxa, which showed the beneficial application value of extruded feed in *Procambarus clarkii* aquaculture.

**Abstract:**

The study was conducted to evaluate the extruded and pelleting feed production on growth performance, intestinal histology and microbiome analysis of juvenile red swamp crayfish, *Procambarus clarkii*. Crayfish were fed either pelleted or extruded feeds that were made using the same formula. Crayfish fed extruded feed had a lower feed conversion ratio, as well as significantly higher levels of trypsin and amylase (*p* < 0.05) than those fed pelleted feed. However, other growth indices and the activity of lipase were not significantly influenced by the feed processing technique (*p* > 0.05). In comparison with the pelleted feed group, the lamina propria thickness of crayfish fed extruded feed was significantly lower (*p* < 0.05). Additionally, the abundance of intestinal microbiota in the extruded feed group was higher than that in the pelleted feed group. The dominant phyla in the intestine of both groups were *Proteobacteria*, *Tenericutes*, and *Firmicutes*, and the relative abundance of *Proteobacteria* in the extruded feed group was significantly higher than that in the pelleted feed group (*p* < 0.05). These results revealed that *P. clarkii* fed extruded feed had higher feed utilization and better intestinal health.

## 1. Introduction

Extrusion is a common feed processing technique that has been increasingly developed and used in aquaculture in recent years [1,2,3]. Compared with pelleted diets, extruded feed increases water stability and durability [4,5], decreases the content of anti-nutritional factors, enhances the utilization of nutrients, and reduces eutrophication in water as a result of lower evacuation of nitrogen and phosphorus [6], all of which are due to the greater levels of heat, moisture and pressure used during the extrusion process [5]. However, high temperature and strong pressure also have some disadvantages, such as destruction of vitamins and dietary enzymes, and extruded feed may be less used in the aquaculture industry because it is technologically difficult to produce and more expensive [5].

Increasing evidence indicates that intestinal microbiota are closely linked to growth, nutrient absorption, and immunity in organisms [7,8,9]. A stable intestinal microbial community can promote growth and disease resistance by balancing digestion and resisting pathogens, and it may even improve breeding efficiency [10]. Thus, the diversity, composition, and functions of microbiota can serve as potential indexes of host health and help predict the influence of dietary additives [11]. Previous studies of aquatic species have revealed significant effects of host genetics, developmental stage, health status, diet, and environmental and geographical factors on the gut microbial community [10,12], and diet has been reported to be a major factor involved in these effects [10]. However, little is known about the relationship between feed processing techniques and intestinal microbiota.

The red swamp crayfish (*Procambarus clarkii*) has become one of the most significant and popular commercial freshwater species in China due to its special food culture, rich nutrition, and delicious taste [13]. The annual aquaculture production of crayfish has increased rapidly in recent years, with the number growing from 852,285 tons in 2016 to 2,393,699 tons in 2020. However, the lack of standard seed and feed processing techniques and increasing problems with disease have made the aquaculture of *P. clarkii* difficult. To improve immunity and increase production to meet the growing demand for this species, researchers have studied its nutrition requirements and feeding management [14,15,16,17]. All of the previous studies of nutritional needs have focused on pelleted feed rather than extruded feed, and the effects of different feed processing techniques on crayfish are poorly understood. The effects of manufacturing processes can vary greatly depending on species [1,2,6,18], and the results of previous studies were not unanimous. Furthermore, developing suitable production technologies to keep feed costs as low as possible is needed [1,2,5].

To address these issues, we subjected feed to two processing techniques to produce extruded and pelleted ones to investigate the effect of processing technique on growth, intestinal histology, and microbiota in *P. clarkii*. Our results can be used to improve the aquaculture of crayfish.

## 2. Materials and Methods

### 2.1. Experimental Animals

*P. clarkii* were provided by the Yangzhong aquaculture base of the Freshwater Fisheries Research Institute of Jiangsu Province (Yangzhong, Jiangsu, China). Crayfish were cultured in an outdoor tank (4 m × 4 m × 4 m, water height: 0.4 m) for 14 days to allow them to acclimate to the experimental environment. All crayfish were fed an equal mixture of the two experimental diets during this period of time.

### 2.2. Experimental Feeds

Table 1 shows the formula of the two experimental feeds, which were produced by Taizhou New Hope Agriculture Co., Ltd. (Taizhou, Jiangsu, China). The ingredients were processed in a melangeur (crushing degree of fineness > 95%, 80-mesh), gradually mixed with water and oil, and then compressed into 2.0 mm pellet size by a pelleted feed machine (IDAH 530, IDAH Machinery Co., Ltd., Taipei, Taiwan) or extruded into the form of extruded feed with the same size by a single-screw extruded feed machine (BULE 125, Bule (Changzhou) Machinery Co., Ltd., Changzhou, Jiangsu, China) at a screw speed was 350 rpm. The in-barrel moisture and modulated temperature were 14.5%, 101 ± 2 °C for the pelleted diet (sinking) and 30–34%, 101 ± 2 °C for the extruded diet (sinking), respectively. After drying, experimental feeds were stored at −20 °C until used.

### 2.3. Experimental Design and Management

One hundred and eighty crayfish juveniles (average body weight: 2.68 ± 0.02 g) were randomly divided into six cement tanks (2 m × 2 m × 2 m, water height: 0.3 m), with each treatment (extruded or pelleted feed) containing three replicates of 30 crayfish (15 male crayfish and 15 female crayfish) per tank. We put a piece of asbestos tile (0.8 m × 0.8 m) with a brick support in the middle of each tank to serve as a shelter to reduce aggressive behaviors among crayfish. Crayfish were fed twice a day (06:30 and 18:30) to satiation and to make sure there were no diet residuals after feeding. The feeding trial was conducted at Yangzhong and lasted for 8 weeks.

The feeding trial was conducted from July to September. A sunshade net above the tanks was used from 11:00 to 16:00 on sunny days. The recirculating aquaculture system was supplied with aerated underground water, which was filtered through an 80-mesh filter at a rate of 5 L min^−1^, and water quality was checked every day. Temperature, salinity, pH, dissolved oxygen content, and total dissolved solid content were checked using an electronic water quality analyzer (model 650MDS, YSI, Yellow Springs, OH, USA), and ammonia, nitrite, and sulfide levels were measured using a W-II water quality analyzer (Octadem, Wuxi, China). Table 2 shows the water quality parameters.

### 2.4. Sampling

After 8 weeks, weights and the number of crayfish were measured and counted after fasting for 24 h. Subsequently, hepatopancreas were collected and weighted to determine physical indices, three crayfish per tank were anesthetized on ice for 15 min, after which the middle part of the intestine (approximately 0.5 cm long) was fixed in 4% paraformaldehyde for histopathology; the rest of the intestine was stored at −20 °C for measurement of digestive enzyme activities. The intestinal contents from 8 crayfish from each tank in each group (24 crayfish per diet group) were randomly collected, and four specimens per replicate were combined to reduce internal individual differences. The samples were frozen in liquid nitrogen and transported to Shanghai Biozeron Biotechnology Co., Ltd. (Shanghai, China) for analysis of the intestinal microbiota.

### 2.5. Measurement Indices and Methods

#### 2.5.1. Feed Proximate Composition

The proximate composition of the feed was determined according to the standard methods of the AOAC (2002) [19]. The moisture content of the feed was determined by weighing the sample, oven-drying the sample to a constant weight at 103 ± 2 °C, and then calculating the percentage of water in the sample. The ash content was determined by weighing the sample, searing the sample in a muffle furnace at 550 °C (SXL-1008; Jing Hong Laboratory Instrument Co., Ltd., Shanghai, China), and then calculating the percentage of ash in the sample. The crude protein content was determined according to the Kjeldahl nitrogen method (K-360; Buchi Labortechnik Ag., Flawil, Switzerland). The crude lipid content was measured using a Soxtherm device (B-811; Buchi Labortechnik Ag.). Vitamin C and vitamin E content were measured by HPLC (E2695; Waters Co., Ltd., Milford, MA, USA). Amino acid contents were measured using an amino acid analyzer (L-8080; Hitachi Ltd., Tokyo, Japan).

#### 2.5.2. Growth Performance and Physical Indices

The growth and physical indices in this study were calculated as follows:Weight gain rate (WGR, %) = 100 × [final weight (g) − initial weight (g)]/initial weight (g)
Specific growth rate (SGR, % day^−1^) = 100 × [ln final weight (g) − ln initial weight (g)]/days of the experiment (d)
Feed intake (FI, g fish^−1^ day^−1^) = total feed intake (g)/number of crayfish/days of the experiment (d)Feed conversion ratio (FCR) = feed intake (g)/weight gain (g)
Survival rate (SR, %) = 100 × final number of crayfish/initial number of crayfish
Hepatosomatic index (HIS, %) = 100 × (hepatopancreas weight/body weight).

#### 2.5.3. Digestive Enzyme Activities

After thawing, the intestinal samples were rinsed with cold distilled water, dried with a paper towel, mixed with four volumes of Tris-HCl buffer (W/V), 50 mM, pH 7.0, and homogenized using an electric homogenizer (IKA T18 digital, ULTRA-TURRAX, Staufen, Germany). Subsequently, homogenates were centrifuged for 10 min at 6000 r min^−1^ at 4 °C. Thereafter, supernatants were used to measure activities of the digestive enzymes, trypsin, amylase, and lipase with kits purchased from Nanjing Jiancheng Bioengineering Institute (Nanjing, China).

#### 2.5.4. Histological Examination

Following, the intestines were fixed in 4% paraformaldehyde for more than 24 h and processed according to standard histological techniques [20]. Afterwards, 4 μm thick tissue slices were obtained by microtome (RM2016, Shanghai Leica Instruments Co., Ltd., Shanghai, China), and half of them were stained with hematoxylin and eosin (H&E) and observed under a Nikon Eclipse CI microscope (Tokyo, Japan). The rest of the tissue sections were stained with a combination of Alcian blue and Periodic Acid-Schiff (PAS) reagent in order to visualize and count goblet cells and lymphocytes. The villus height, mucosal fold width, lamina propria thickness, and intestinal wall thickness in the intestine were analyzed and measured using Case Viewer 2.4 (3DHISTECH, Budapest, Hungary) and Image-Pro Plus 6.0 (Media Cybernetics, Rockville, MD, USA).

#### 2.5.5. PCR Intestinal Microbiota Analysis

Total DNA was extracted from intestine tissues using the E.Z.N.A. ^®^DNA Kit (Omega Bio-Tek, Norcross, GA, USA). Common primers (Table 3) were used to amplify the 16S rRNA genes of V4–V5 regions. The PCR amplicons were isolated from agarose gels (2%), purified using the AxyPrep DNA Gel Extraction Kit (Axygen Biosciences, Union City, CA, USA), and then quantified. Sequencing was conducted using an Illumina MiSeq platform (Shanghai Biozeron). The raw reads were uploaded to the NCBI SRA database (SRP302469).

### 2.6. Statistical Analysis

The alpha diversity (Chao 1, Shannon, and Simpson) was determined using Mothur v.1.21.1 and the beta diversity was measured via principal coordinate analysis (PCoA), which was performed using UniFrac. Venn diagrams were created using the online tool “Draw Venn Diagram” (http://bioinf-ormatics.psb.ugent.be/webtools/Venn (accessed on 16 November 2020)). Diagrams of microbial communities were drawn by Origin 8.0 software. Test data were processed in Excel, then the R (version 3.5.0, https://www.r-project.org/ (accessed on 24 April 2022)) function *aov()* was employed to statistical analysis of growth performance, digestive enzyme activities, histological indices and intestinal microbiota, the significance of difference between two groups was analyzed by T-test and least square difference (LSD). All data are presented as mean ± SE.

## 3. Results

### 3.1. Growth Performance

Crayfish fed the extruded feed had lower FCR than those fed pelleted feed (*p* < 0.05), but no significant differences in FBW, WGR, SGR, SR, FI, and HSI were found between the two diets (*p* > 0.05) (Table 4).

### 3.2. Digestive Enzymes Activities

The activities of trypsin and amylase were higher in the extruded feed group than in the pelleted feed group (*p* < 0.05), but the activity of lipase did not differ significantly between the two feed types (*p* > 0.05) (Table 5).

### 3.3. Histology of the Intestine

Table 6 and Figure 1 show the results of the histomorphological analysis of the intestine of crayfish fed pelleted and extruded feeds. Compared with the pelleted feed group, the lamina propria thickness was significantly lower in crayfish fed extruded feed (*p* < 0.05). The latter also had slightly smaller villus height, mucosal fold width, and intestinal wall thickness, but the differences were not significant (*p* > 0.05). The numbers of goblet cells and lymphocytes were small in crayfish from both groups, and no significant visible differences in these cell types in the intestine were detected between the two experimental diet groups (*p* > 0.05).

### 3.4. Intestinal Microbiota

In total, 721,291 high-quality sequences were obtained from the intestinal contents of 12 samples, ranging from 50,411 to 72,853. Table 7 shows the alpha diversity of the samples. According to the results of the Chao1 index, the abundance of intestinal microbiota was lower in the pelleted feed group than in the extruded feed group (*p* < 0.05). The Shannon and Simpson indexes were not significantly influenced by feed type (*p* > 0.05).

The PCoA analysis of the community structures of the intestinal microbiota revealed that the communities from the pelleted feed group were distinguished from those from the extruded feed group on principal coordinate axis 1, which reflected a large variation (49.67%) (Figure 2a). In total, 121 and 256 operational taxonomic units (OTUs) were identified only in the pelleted feed group or in the extruded feed group, respectively; 496 overlapping OTUs were identified in both diets (Figure 2b).

Figure 3 shows the core microflora communities in the two feed groups at the phylum and genus levels. The dominant phyla in the intestinal microbiota of crayfish fed pelleted feed were *Proteobacteria* (67.32%), *Tenericutes* (13.05%), and *Firmicutes* (8.43%). The dominant phyla in the intestinal microbiota of crayfish fed extruded feed were *Proteobacteria* (72.34%), *Firmicutes* (9.09%), and *Tenericutes* (6.28%). The population percentages of other microbe species were <5%. The relative abundance of *Proteobacteria* in crayfish fed extruded feed was significantly higher than that of crayfish fed pelleted feed (*p* < 0.05). Of the 343 observed genera, three were predominant (relative abundance > 5%) in the intestinal microbiota of crayfish in the extruded feed group (*Citrobacter* (42.28%), *Candidatus bacilloplasma* (8.75%), and *Aeromonas* (7.35%)) and four were predominant in the pelleted feed group (*Citrobacter* (33.19%), *Rhodobacter* (7.39%), *(Anaerorhabdus) furcosa* group (6.19%), and *Candidatus bacilloplasma* (5.84%)). The experimental diet had no significant effect on them (*p* > 0.05).

## 4. Discussion

Feed processing technology has received increasing attention in recent years. In most of these studies, feed efficiency was significantly improved when aquatic animals were fed extruded feed [18,21]. However, the effects of feed processing on growth performance were not unanimous. Several researchers reported a reduction of growth in organisms fed pelleted feed compared to extruded feed [2,6,18], which was in contrast with [2], who found that growth was unaffected when channel catfish (*Ictalurus punctatus*) were fed feed processed by extraction or extrusion.

Our results showed that *P. clarkii* fed the extruded feed had a better FCR than those fed pelleted feed (Table 4). This result likely was related to the high gelatinization that resulted from extrusion, which increased the utilization of starch and improved the water-borne durability and nutrient intactness of the feed [22]. The better FCR of the extruded feed also may be related to the compaction and retention of nutrients, which would minimize the amount of leaching into the water. This feature would lead to higher gut nutrient content and digestion, longer gastric evacuation time, and slower emptying rates of the gastric intestinal tract compared to that in fish fed pelleted feed (fish meal content was 6%, 0%, and 27%, respectively) [2,18,23]. Furthermore, based on the research previously (fish meal content was 6%) [2], the diet formulation in this study could meet the nutritional needs of crayfish, thus the improvement of nutrient digestibility and utilization rate by extruded processing wasn’t extremely obvious [24]. Although the feeding economies of the two feeds differed significantly, growth performance of crayfish was not compromised by the feed processing techniques tested in this study.

The utilization of nutrients by aquatic animals is determined largely by the level of digestive enzyme activity in the intestine, which can be affected by exogenous and endogenous factors. Diet is one of the major exogenous factors [25]. Modulation of digestive enzyme activity by feed type (e.g., pelleted feed and extruded feed) in a variety of aquatic species has been reviewed recently: the latter led to higher levels of digestive enzyme activity in some fish [18,26]. Enzyme release was probably related to gastric evacuation time, which was extended in the extruded feed group. In agreement with results of previous studies, we also found that higher levels of intestinal trypsin and amylase activity occurred in crayfish fed extruded feed compared to pelleted feed (Table 5). These findings confirmed the beneficial effects of extruded feed on intestinal digestion processes and showed that increased digestive enzyme activity improved the FCR in the crayfish fed extruded pellets.

Improvements in intestinal histology are beneficial to feed utilization and health status because they increase the organism’s ability to prevent bacterial infection of the mucosal epithelium [27]. The enteric and absorptive capacity of the intestine was closely linked to villus height, mucosal fold height, and mucosa thickness [28]. Moreover, goblet cells have significant effects on the digestion and health of aquatic animals by synthesizing and secreting mucins, which lubricate and protect the intestinal epithelium [29,30,31]. However, little is known about the effects of feed processing procedures on intestinal morphology. In this study, we found that the lamina propria thickness in crayfish fed pelleted feed was significantly increased compared to that of the extruded feed group (Table 6). Extended intestinal lamina propria were also founded in intestinal damage induced by nutritional stress [20]. Although the feed processing method did not significantly affect the numbers of goblet cells and lymphocytes and no pathomorphological changes were detected between crayfish fed the two diets (Figure 1). Similarly, feed processing technique and size did not significantly affect histomorphology of digestive organs in fish [32,33].

Intestinal microbiota is closely related to growth, feed utilization, digestion, and nutrition absorption of aquatic species [34,35,36]. Moreover, a stable and beneficial microbial composition has a favorable effect on immunity and health of the host [18,37]. Recent studies indicated that the variation of intestinal microbiota is more affected by host development and diet than by the geography and surrounding environment [10]. Therefore, numerous studies have focused on the effects of diet on microbial composition [11,38].

To the best of our knowledge, no prior study has reported the effect of feed processing technique on intestinal microbiota for any aquatic animal. We found no significant difference in diversity (reflected by the Shannon and Simpson indexes) of crayfish, but the abundance of intestinal microbiota in the pelleted feed group was lower than that in the extruded feed group (Table 7). The lower FCR observed in the extruded feed group might be related to the higher abundance of intestinal microbiota in this group, which could improve nutrient absorption [39]. In accordance with the Chao1 index results, the extruded feed group had more OTUs than the pelleted feed group (Figure 2), further confirming that the microbial composition was richer in crayfish fed this diet [20]. The lower number of OTUs in crayfish fed pelleted feed suggested that these crayfish may ingest fewer nutrients due to leaching from the feed prior to ingestion [40]. The PCoA plot showed that the intestinal microbiota of the two groups gathered into two independent clusters, which indicated that the feed processing technique altered the intestinal microbiota structure of the crayfish. Previous studies also reported that a flora imbalance caused by diet might be responsible for the variation in the structure of intestinal microbiota [41,42]. Thus, changes of the intestinal microbial flora induced by a specific feed processing technique might have negative impacts on crayfish.

In our study (Figure 3), *Proteobacteria*, *Tenericutes*, *Firmicutes*, and *Bacteroidetes* were the dominant phyla in both groups. Previous studies reported that 90% of intestinal microbiota in many aquatic animals belong to *Proteobacteria*, *Bacteroidetes*, and *Firmicutes* [43,44,45], which indicates that these bacteria are closely associated with crucial functions of the intestine, including nutrient absorption, digestion, and immunity [46]. *Proteobacteria* was the most dominant phylum in all crayfish examined in our study, which agrees with findings of previous studies of the intestinal microbiota of healthy *P. clarkii* [10,12,40] and other crustaceans, including various shrimp species [47,48,49,50]. Furthermore, the relative abundance of *Proteobacteria* was significantly higher in crayfish fed the extruded feed diet than in those fed the pelleted feed. *Proteobacteria* has also been identified as one of the most abundant phyla in shrimp species investigated, and this phylum plays a critical part in various biochemical functions, such as carbon and nitrogen cycling [20]. Studies have also indicated that *Proteobacteria* is likely involved in some biogeochemical processes and in the intestine of crustaceans [51]. High abundance of *Proteobacteria* may also pose potential risks for hosts, as higher numbers of *Proteobacteria* were observed in shrimp infected with pathogenic bacteria or in those with poor growth [52,53].

At the genus level, Illumina high-throughput sequencing results demonstrated that the dominant genera were not consistent between the two feed groups. However, *Citrobacter* was the most abundant genus in crayfish fed both diets. *Citrobacter rodentium* is an attaching and effacing bacterial pathogen that shares pathogenic mechanisms with enteropathogenic and enterohaemorrhagic *Escherichia coli*, which can cause various diarrheal diseases and death [54,55,56]. In our study, the relative abundance of *Citrobacter* in crayfish fed pelleted feed was higher than that in specimens fed with extruded feed, although the difference was not statistically significant. No significant differences in abundances of other dominant bacteria were detected.

## 5. Conclusions

In spite of short experimental duration, the results clearly demonstrated that extruded feed could improve feed utilization and nutrition absorption by increasing digestive enzymes activities and stabilizing intestinal morphology. Furthermore, crayfish fed with extruded feed also enhanced the richness of intentional microbiota and reduced the proportion of harmful microbial taxa. In spite of further research focusing on this mechanism being necessary, it could be concluded that the extruded feed showed its beneficial application value in *Procambarus clarkii* aquaculture.

## Figures and Tables

**Figure 1 animals-12-02252-f001:**
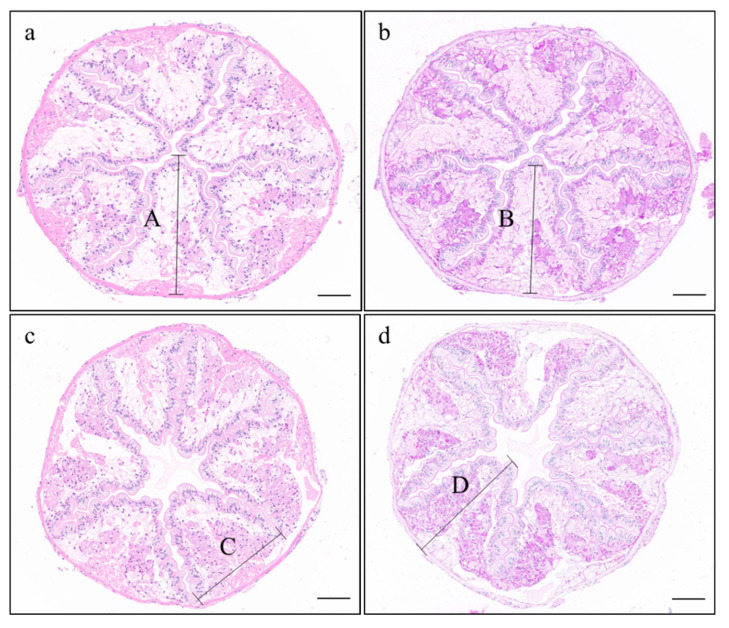
Histological sections of the intestine of *P. clarkii*. (**a**) pelleted feed group, H&E stain (10×); (**b**) pelleted feed group, PAS stain (10×); (**c**) extruded feed group, H&E stain (10×); (**d**) extruded feed group, PAS stain (10×). Scale bar: 100 μm. Uppercase letters indicate examples of measurements of villus height, mucosal fold width, lamina propria thickness and intestinal wall thickness (A, B, C and D, respectively).

**Figure 2 animals-12-02252-f002:**
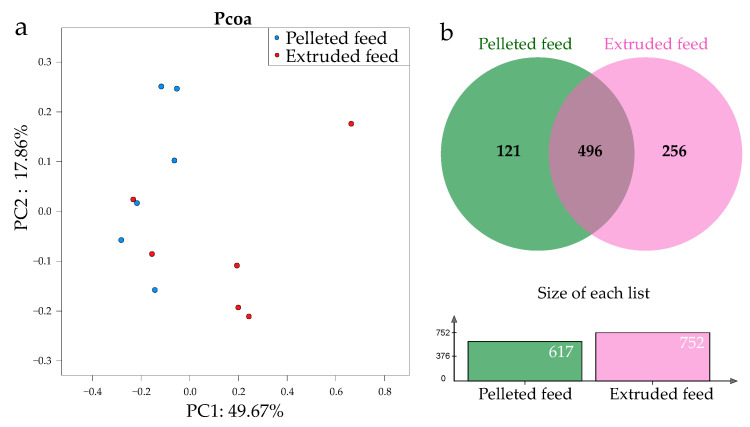
(**a**) Principal coordinate analysis (PCoA) of intestinal bacterial communities of *P. clarkii* fed pelleted feed and extruded feed; based on weighted UniFrac distances, each point represents a sample, and different colors represent different diets. (**b**) Venn diagram showing shared and unique OTUs of *P. clarkii* fed different diets.

**Figure 3 animals-12-02252-f003:**
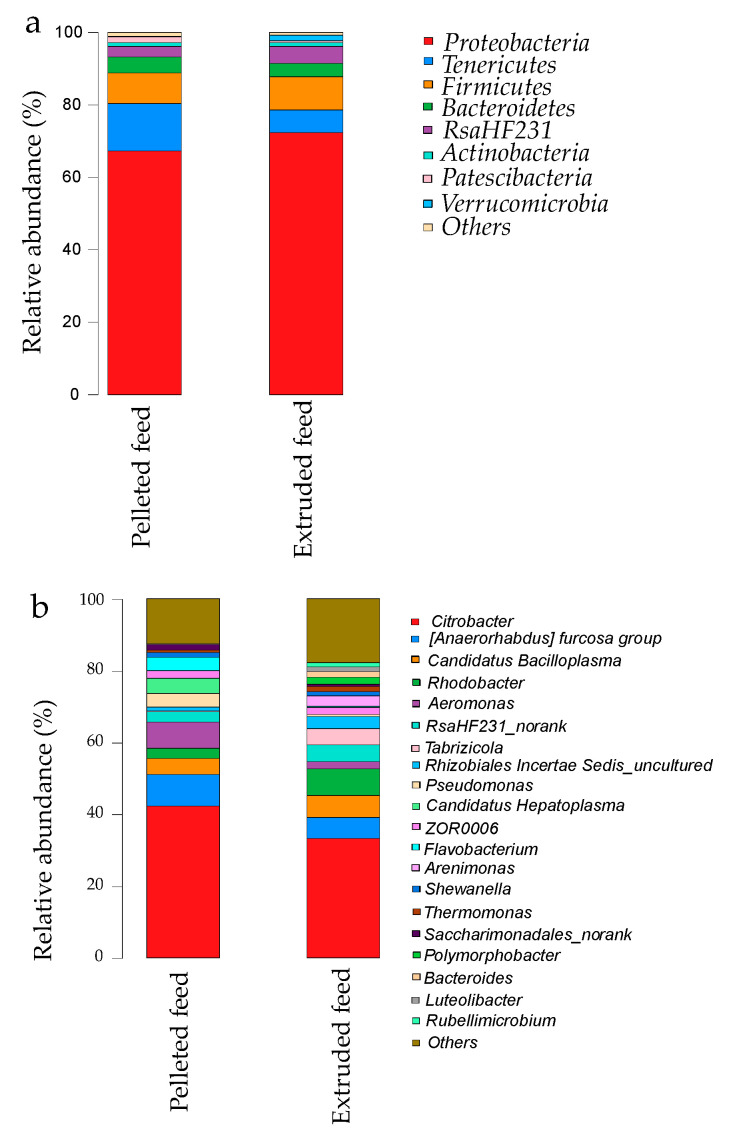
Relative abundances of the dominant bacteria at the (**a**) phylum level and (**b**) genus level in *P. clarkii* fed pelleted feed and extruded feed.

**Table 1 animals-12-02252-t001:** Ingredients and proximate composition of experimental feeds (g kg^−1^).

Ingredients ^a^	Pelleted Feed	Extruded Feed
Soybean meal	230.0	230.0
Rapeseed meal (green)	160.0	160.0
Rapeseed meal	70.0	70.0
DDGS	70.0	70.0
Blood globulin powder	20.0	20.0
Rice bran	70.0	70.0
Wheat middling	250.0	250.0
Distilled grain	81.5	81.5
Choline chloride 60%	5.0	5.0
Soybean oil	15.0	15.0
Calcium biphosphate	18.0	18.0
Vitamin premix ^b^	3.0	3.0
Mineral premix ^c^	5.0	5.0
Antiseptic	1.5	1.5
Ethoxyquin	1.0	1.0
Total	1000.0	1000.0
Proximate composition (as feed)		
Dry matter (DM)	895.0	901.0
Crude protein	315.9	307.9
Crude lipid	55.0	51.0
Ash	114.0	121.0
Vitamin E (IU/kg)	102.0	97.7
Lysine (Lys)	17.3	18.2
Methionine (Met)	2.8	2.7
Threonine (The)	12.3	11.8
Arginine (Arg)	17.6	16.4
Total amino acids (TAA)	267.3	260.0

Note: ^a^ Ingredients were purchased from the Taizhou New Hope Agriculture Co., Ltd. (Taizhou, Jiangsu, China), and the protein contents are as follows: soybean meal (456 g kg^−1^), rapeseed meal (green) (350 g kg^−1^), rapeseed meal (361 g kg^−1^), DDGS (252 g kg^−1^), blood globulin powder (900 g kg^−1^), rice bran (133 g kg^−1^). ^b^ Vitamin premix (mg or IU kg^−1^ diet): vitamin A, 8000 IU; vitamin E, 150 mg; vitamin K, 50 mg; thiamine, 80 mg; riboflavin, 50 mg; niacin, 150 mg; pantothenic acid, 150 mg; pyridoxine HCl, 50 mg; biotin, 1 mg; cyanocobalamin, 0.02 mg; folic acid, 10 mg; ascorbic acid, 300 mg; vitamin D3, 2000 IU. ^c^ Mineral premix (mg or g kg^−1^ diet): copper sulphate, 2.0 g; iron sulphate, 25 g; zinc sulphate, 22 g; manganese sulphate, 7 g; sodium selenite, 0.04 g; potassium iodide, 0.026 g; cobalt chloride, 0.1 g.

**Table 2 animals-12-02252-t002:** Water quality parameters.

Parameters	Pelleted Feed	Extruded Feed
Temperature (°C)	20–27	20–27
Dissolved oxygen (mg/L)	7.81–9.02	7.75–8.99
pH	7.60–8.52	7.51–8.33
Total dissolved solids (g/L)	0.198–0.215	0.194–0.215
Salinity (%)	0.14–0.16	0.14–0.16
Ammonia (mg/L)	0.024–0.035	0.023–0.033
Nitrite (mg/L)	0.027–0.033	0.027–0.033
Sulfide (mg/L)	0.001–0.002	0.001–0.002

**Table 3 animals-12-02252-t003:** Forward (F) and reverse (R) primers were used for PCR-test.

Name	Primer Sequence (5′-3′)	Accesion Number
LinA_341F	CCTAYGGGRBGCASCAG	NR024570.1
LinB_806R	GGACTACNNGGGTATCTAAT	

**Table 4 animals-12-02252-t004:** Effects of pelleted and extruded feeds on growth performance and physical indices of *P. clarkii*.

Indices	Pelleted Feed	Extruded Feed
IBW (g)	2.70 ± 0.01	2.67 ± 0.03
FBW (g)	21.54 ± 0.81	21.50 ± 2.04
WGR (%)	698.98 ± 27.51	705.40 ± 72.10
SGR (% day^−1^)	3.85 ± 0.06	3.86 ± 0.17
FI (g fish^−1^ day^−1^)	17.84 ± 2.57	14.59 ± 1.67
FCR	0.94 ± 0.09 ^a^	0.78 ± 0.06 ^b^
SR (%)	78.89 ± 5.67	82.22 ± 6.85
HSI (%)	6.82 ± 0.43	7.05 ± 0.52

Note: Values are means ± SE. In the same row, values with different superscript letters indicate a significant difference (^a^, ^b^; *p* < 0.05). IBM: initial body weight (g). FBM: final body weight (g).

**Table 5 animals-12-02252-t005:** Effects of pelleted and extruded feeds on digestive enzyme activities in the intestine of *P. clarkii*.

Group	Trypsin (U/mgprot)	Lipase (U/mgprot)	Amylase (U/mgprot)
Pelleted feed	319.49 ± 20.17 ^b^	0.0013 ± 0.00	1.52 ± 0.13 ^b^
Extruded feed	398.42 ± 30.07 ^a^	0.0011± 0.00	2.23 ± 0.18 ^a^

Note: Values are means ± SE. In the same column, values with different superscript letters indicate a significant difference (^a^, ^b^; *p* < 0.05).

**Table 6 animals-12-02252-t006:** Effects of pelleted and extruded feeds on intestinal microscopic structure of *P. clarkii*.

Group	Villus Height (mm)	Mucosal Fold Width (mm)	Lamina Propria Thickness (mm)	Intestinal Wall Thickness (mm)
Pelleted feed	381.45 ± 5.39	354.58 ± 14.21	345.62 ± 2.92 ^a^	392.52 ± 8.11
Extruded feed	376.19 ± 3.45	324.43 ± 72.05	329.80 ± 2.01 ^b^	391.33 ± 5.84

Note: Values are means ± SE. In the same column, values with different superscript letters indicate a significant difference (^a^, ^b^; *p* < 0.05).

**Table 7 animals-12-02252-t007:** Number of reads, statistical estimated community richness index (Chao1), community diversity indexes (Shannon and Simpson), and Good’s coverage for 16S rRNA libraries of *P. clarkii* intestinal microbial ecosystems.

Group	Number of Seqs	Read Number	Chao1	Shannon	Simpson	Good’s Coverage
Pelleted feed	50411–71402	49862–70676	419.47 ± 44.88 ^b^	3.04 ± 0.39	0.11 ± 0.06	99.86 ± 0.02
Extruded feed	52454–72853	51950–72095	533.73 ± 76.05 ^a^	3.54 ± 0.51	0.07 ± 0.04	99.86 ± 0.02

Note: Values are means ± SE. In the same column, values with different superscript letters indicate a significant difference (^a^, ^b^; *p* < 0.05).

## Data Availability

The datasets generated or analysed during the current study are available in manuscript and NCBI SRA database (SRP302469).

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
