# Peer review of "Effects of Pelleted and Extruded Feed on Growth Performance, Intestinal Histology and Microbiota of Juvenile Red Swamp Crayfish (Procambarus clarkii)"

_animals, 2022, doi:10.3390/ani12172252_

Round 1

Reviewer 1 Report

Please remove from the abstract the (p < 0.05)

Line 42: correct the reference after “enzymes”

Line 53-55: to better link the paragraphs, I would start with this sentence and then start with the relation diet-microbiota.

Line 71: “extruded and pelleted ones”

Line 72: “clarkii”

Line 194: please report p < 0.05 and remove P > 0.05. Check along the text.

Histological analyses: figures do not report a visible scale bar, please add it. In addition, a lot of measurements were done on the epithelium, and, for that reason, authors should add an additional figure in which they show how they take these measurements. Please highlight with arrows each measurement. The “intestinal wall thickness” is not clear. Finally, how do you measure the number of goblet cells or lymphocytes? There are no numeric results for these two parameters. Please report the count of gc and lymphocytes or convert it into an index.

Lines 287-281: Here the authors refer to fish but the gastrointestinal tract and the digestive processes of fish are different from those of crustaceans.

Line 282: “XX et al [2]”

Line 311: they are not fish

Line 312: the absence of an inflammatory response should be demonstrated by pictures at higher magnification focused on the submucosa width. The authors found a slight but significant increase in the lamina propria width, but there are no elements to clearly confirm the absence of an inflammatory influx. In this sense, the presence of clearer pictures and an inflammatory influx index can help. I leave a recent paper that can help author to understand what I mean http://dx.doi.org/10.1016/j.aquaculture.2022.738137

Author Response

We are truly grateful to your critical comments and thoughtful suggestions. We have addressed the comments and revised the manuscript. Point by point responses to your comments are listed below this letter.

Reviewer-1's comments

Please remove from the abstract the (p < 0.05)

Reply: this part has been revised according to your suggestion.

Line 42: correct the reference after “enzymes”

Reply: “enzymes1” has been corrected into “enzymes”.

Line 53-55: to better link the paragraphs, I would start with this sentence and then start with the relation diet-microbiota.

Reply: thank you for your suggestion. We think it would be better to emphasize the significance of this study if we put the sentence at the end of the paragraph.

Line 71: “extruded and pelleted ones”

Reply: “extruded feed and pelleted feed” has been changed into “extruded and pelleted ones”.

Line 72: “clarkii”

Reply: “clarkia” has been corrected into “clarkii

Line 194: please report p < 0.05 and remove P > 0.05. Check along the text.

Reply: “P > 0.05” has been corrected into “p < 0.05”.

Histological analyses: figures do not report a visible scale bar, please add it. In addition, a lot of measurements were done on the epithelium, and, for that reason, authors should add an additional figure in which they show how they take these measurements. Please highlight with arrows each measurement. The “intestinal wall thickness” is not clear. Finally, how do you measure the number of goblet cells or lymphocytes? There are no numeric results for these two parameters. Please report the count of gc and lymphocytes or convert it into an index.

Reply: the scale bar and arrows have been added in the revised manuscript.

All measurements can be found with arrows in each figures.

The “intestinal wall thickness” has been showed and highlighted with arrows in the figures.

The goblet cells and lymphocytes were hardly found, therefore they weren’t counted.

Lines 287-281: Here the authors refer to fish but the gastrointestinal tract and the digestive processes of fish are different from those of crustaceans.

Reply: the research of caspian roach has been replaced with crayfish, however, some studies on fish has to be cited because of the limited information currently available on the utilization of extruded feed on crustaceans.

Line 282: “XX et al [2]”

Reply: the sentence has been changed into “Furthermore, based on the research previously [2] …”.

Line 311: they are not fish

Reply: “fish” has been corrected into “crayfish”.

Line 312: the absence of an inflammatory response should be demonstrated by pictures at higher magnification focused on the submucosa width. The authors found a slight but significant increase in the lamina propria width, but there are no elements to clearly confirm the absence of an inflammatory influx. In this sense, the presence of clearer pictures and an inflammatory influx index can help. I leave a recent paper that can help author to understand what I mean http://dx.doi.org/10.1016/j.aquaculture.2022.738137

Reply: the sentence “this lack of negative effects demonstrated that the two feed processing techniques did not promote an inflammatory response in the intestine” has been deleted.

 Once again, thank you very much for your comments and suggestions.

Reviewer 2 Report

Review report: Manuscript ID animals:1818812

·         The present MS is very important to the science of aquaculture nutrition as it answers some of the burning issues on effects of feed processing to targeted cultured animals. The studies on pellets and extruded feed comparison have been conducted before with mixed outcomes, therefore, there is less novelty in this present study except in the variables histology and microbiota which have not been thoroughly investigated in similar prior studies.

·         I have the following thorny issues that must be addressed before re-consideration:

Language and writing

·         The authors must re-write the manuscript holistically, engage a native English writer and resubmit only when the English has been improved. I would suggest getting assistance form English editing company elsewhere or if resources permit get assistance from experts in Animals journal.

Title

·         Must include the English name for the fish before the scientific name as follows Red swamp crayfish procamburus clarkii

Summary

·         The summary is just a summary not simple summary therefore delete the word simple. The summary must be re written and it must capture the main gap or problem that prompted the research study as solution to the gap or problem stated. The brief research design must be outlined and that includes feeds formulated and the treatments used, the duration of the study, the short description of results and brief discussion of outcomes as well as the overall significance of the study especially how it addresses global issues in aqua feed processing and its contribution to the existing body of aquaculture feed processing knowledge. As it is, the present summary is empty since it is void of important information to the reader. For better reference and guidance on summary  refer to a published article below:

Ø  Mzengereza, K.; Ishikawa, M.; Koshio, S.; Yokoyama, S.; Yukun, Z.; Shadrack, R.S.; Seo, S.; Kotani, T.; Dossou, S.; Basuini, M.F.E.; Dawood, M.A.O. Growth Performance, Growth-Related Genes, Digestibility, Digestive Enzyme Activity, Immune and Stress Responses of de novo Camelina Meal in Diets of Red Seabream (Pagrus major). Animals 2021, 11, 3118. https://doi.org/10.3390/ani11113118

Abstract

·         Line 19-20 should change to the study was conducted to evaluate the extruded and pelleting feed production on growth performance, intestinal morphology/histology and microbiome analysis of juvenile red swamp crayfish,

 Material and methods

·         The duration of the experiment was only 2 months. The effect of the experimental diets on fishes could take more time. Usually, these kinds of experiments last for at least 4 months.

·         The statistical analysis using Analysis of Variance (ANOVA) has assumptions one of which is that the factors or treatments in this case feed processing method must be aleast three. I think Use of ANOVA in this experiment was a violation to the assumptions of ANOVA that must always be met, l would recommend use of T-test to compare only two treatments *two processing methods i. epelleting and extruder).

·         In statistical analysis please indicate which adhoc analysis you used to separate significantly different treatment means e. g Duncan multiple range, Turkey test, Least Square Difference (LSD), which one did you use??

·          Sub heading 2.5.5. Intestinal microbiota analysis should change to PCR Intestinal microbiota analysis. The PCR-test must be Indicated in the Sub -heading

·         The primers used in the PCR Intestinal microbiota analysis must be contained in a table and well labeled showing primer, accession number, primer sequence. Refer to the published article below on presentation of primers in literature

Ø  Mzengereza, K.; Ishikawa, M.; Koshio, S.; Yokoyama, S.; Yukun, Z.; Shadrack, R.S.; Seo, S.; Kotani, T.; Dossou, S.; Basuini, M.F.E.; Dawood, M.A.O. Growth Performance, Growth-Related Genes, Digestibility, Digestive Enzyme Activity, Immune and Stress Responses of de novo Camelina Meal in Diets of Red Seabream (Pagrus major). Animals 2021, 11, 3118. https://doi.org/10.3390/ani11113118

·         Line 181-185 subsequently, the alpha diversity (Chao 1, Shannon, and Simpson) was determined 181 using Mothur v.1.21.1 and the beta diversity was measured via principal coordinate anal- 182

ysis (PCoA), which was performed using UniFrac. Venn diagrams were created using the 183

online tool “Draw Venn Diagram” (http://bioinf-ormatics.psb.ugent.be/webtools/Venn). 184

Diagrams of microbial communities were drawn by Origin 8.0 software must be removed form General methods and material and must be put on the section of statistical analysis.

Results

·         In Table 3 for growth performance, it is important to calculate Feed Intake FI, (g/fish/duration of study: days) = (total feed intake (g) / number of fishes) in a feeding period. FI is an important indicator for palatability and subsequent ingestion of the diet. That would reveal the impact of processing methods to the flavour and acceptability of the final product.

·         In Table 3 for growth performance, it is important to calculate HSI, Hepatosomatic index = 100× (liver weight/body weight) to reflect the liver function, since you did not analyze liver enzymes or blood parameters, therefore HIS becomes handy to explain results on health aspects. Fish processing methods is poised to have an impact on liver cells which must be assessed to see the extent of the effects.

·         Table 4 there are two different Units for enzymes (U/gprot and Umgprot). Please use uniform Units for all three enzymes.

·         In all tables l suggest you change to use alphabetical letter (a, b, c etc) as superscripts to show adhoc analysis results separating significantly different means in place of Asterix (*) that is currently used.

Discussion

·         Put the subheading for the discussion section like results. Also, please keep a sequence in subheading for investigated factors, in material and method, result, and discussion.

·         As a general comment: the size of fish and the fish meal content in the diets play a crucial role in fish processing. When you compare your results with other studies, to avoid any misconception, please add the fish meal contents of diets in their studies

·         Some parts of the discussion are better updated with research in 2021 and 2020 as they refer to some old references. Please update the discussion with the latest studies as much as possible.

·         Please clarify which fish species throughout the discussion when you say “fish”.

·         Line 321-322 to the best of our knowledge, no prior study has reported the effect of feed pro- cessing technique on intestinal microbiota for any aquatic animal. We found no significant difference in diversity (reflected by the Shannon and Simpson indexes) of crayfish.

Comment: Include Table number or Figure number where that data is presented

Conclusion

·         Finally, the article must address the wider relevance of your results, which must be easily understood and summarized in conclusions. Conclude based on the level of the treatment.

Decision

·         My decision on the manuscript is that it be considered after major revision.

Author Response

We are truly grateful to your critical comments and thoughtful suggestions. We have addressed the comments and revised the manuscript. Point by point responses to your comments are listed below this letter.

Reviewer-2's comments

The present MS is very important to the science of aquaculture nutrition as it answers some of the burning issues on effects of feed processing to targeted cultured animals. The studies on pellets and extruded feed comparison have been conducted before with mixed outcomes, therefore, there is less novelty in this present study except in the variables histology and microbiota which have not been thoroughly investigated in similar prior studies.

I have the following thorny issues that must be addressed before re-consideration:

Language and writing

The authors must re-write the manuscript holistically, engage a native English writer and resubmit only when the English has been improved. I would suggest getting assistance form English editing company elsewhere or if resources permit get assistance from experts in Animals journal.

Reply: the language has been improved by an English editing company — International Science Editing (http://www.internationalscienceediting.com), and the contribution of their revision has been mentioned in the acknowledgments section. Furthermore, the manuscript can be revised again based on journal requirements.

Title

Must include the English name for the fish before the scientific name as follows Red swamp crayfish procamburus clarkii

Reply: the title has been changed into “Effects of pelleted and extruded feed on growth performance, intestinal histology and microbiota of juvenile Red swamp crayfish (Procambarus clarkii)”.

Summary

The summary is just a summary not simple summary therefore delete the word simple. The summary must be re written and it must capture the main gap or problem that prompted the research study as solution to the gap or problem stated. The brief research design must be outlined and that includes feeds formulated and the treatments used, the duration of the study, the short description of results and brief discussion of outcomes as well as the overall significance of the study especially how it addresses global issues in aqua feed processing and its contribution to the existing body of aquaculture feed processing knowledge. As it is, the present summary is empty since it is void of important information to the reader. For better reference and guidance on summary refer to a published article below:

Ø  Mzengereza, K.; Ishikawa, M.; Koshio, S.; Yokoyama, S.; Yukun, Z.; Shadrack, R.S.; Seo, S.; Kotani, T.; Dossou, S.; Basuini, M.F.E.; Dawood, M.A.O. Growth Performance, Growth-Related Genes, Digestibility, Digestive Enzyme Activity, Immune and Stress Responses of de novo Camelina Meal in Diets of Red Seabream (Pagrus major). Animals 2021, 11, 3118. https://doi.org/10.3390/ani11113118

Reply: this part has been re-written according to your suggestion. However, I think the title of “simple summary” is a fixed format of the journal, I can certainly delete the word simple if it’s not necessary.

Abstract

Line 19-20 should change to the study was conducted to evaluate the extruded and pelleting feed production on growth performance, intestinal morphology/histology and microbiome analysis of juvenile red swamp crayfish,

Reply: this part has been revised according to your suggestion.

Material and methods:

The duration of the experiment was only 2 months. The effect of the experimental diets on fishes could take more time. Usually, these kinds of experiments last for at least 4 months.

Reply: considering the actual culture of crayfish were normally lasted for two months, thus the feeding trial was conducted for 8 weeks. Certainly, we will try to extend the duration of the experiment in the further study based on your suggestion.

The statistical analysis using Analysis of Variance (ANOVA) has assumptions one of which is that the factors or treatments in this case feed processing method must be at least three. I think Use of ANOVA in this experiment was a violation to the assumptions of ANOVA that must always be met, l would recommend use of T-test to compare only two treatments *two processing methods i. epelleting and extruder.

In statistical analysis please indicate which ad-hoc analysis you used to separate significantly different treatment means e. g Duncan multiple range, Turkey test, Least Square Difference (LSD), which one did you use??

Reply: statistical analysis has been revised according to your request, and the changes of results caused by analysis methods are also revised in the manuscript.

Sub heading 2.5.5. Intestinal microbiota analysis should change to PCR Intestinal microbiota analysis. The PCR-test must be Indicated in the Sub -heading.

Reply: the sub heading “Intestinal microbiota analysis” has been replaced with “PCR intestinal microbiota analysis”.

The primers used in the PCR Intestinal microbiota analysis must be contained in a table and well labeled showing primer, accession number, primer sequence. Refer to the published article below on presentation of primers in literature

Ø  Mzengereza, K.; Ishikawa, M.; Koshio, S.; Yokoyama, S.; Yukun, Z.; Shadrack, R.S.; Seo, S.; Kotani, T.; Dossou, S.; Basuini, M.F.E.; Dawood, M.A.O. Growth Performance, Growth-Related Genes, Digestibility, Digestive Enzyme Activity, Immune and Stress Responses of de novo Camelina Meal in Diets of Red Seabream (Pagrus major). Animals 2021, 11, 3118. https://doi.org/10.3390/ani11113118

Reply: the table of primers (table 3) has been added.

Line 181-185 subsequently, the alpha diversity (Chao 1, Shannon, and Simpson) was determined 181 using Mothur v.1.21.1 and the beta diversity was measured via principal coordinate anal- 182 ysis (PCoA), which was performed using UniFrac. Venn diagrams were created using the 183 online tool “Draw Venn Diagram” (http://bioinf-ormatics.psb.ugent.be/webtools/Venn). 184 Diagrams of microbial communities were drawn by Origin 8.0 software must be removed form General methods and material and must be put on the section of statistical analysis.

Reply: this part has been revised according to your request.

Results

In Table 3 for growth performance, it is important to calculate Feed Intake FI, (g/fish/duration of study: days) = (total feed intake (g) / number of fishes) in a feeding period. FI is an important indicator for palatability and subsequent ingestion of the diet. That would reveal the impact of processing methods to the flavour and acceptability of the final product.

Reply: information of feed intake has been added in the revised manuscript.

In Table 3 for growth performance, it is important to calculate HSI, Hepatosomatic index = 100× (liver weight/body weight) to reflect the liver function, since you did not analyze liver enzymes or blood parameters, therefore HIS becomes handy to explain results on health aspects. Fish processing methods is poised to have an impact on liver cells which must be assessed to see the extent of the effects.

Reply: information of hepatosomatic index has been added in the revised manuscript.

Table 4 there are two different Units for enzymes (U/gprot and Umgprot). Please use uniform Units for all three enzymes.

Reply: U/gprot were changed into U/mgprot.

In all tables l suggest you change to use alphabetical letter (a, b, c etc) as superscripts to show adhoc analysis results separating significantly different means in place of Asterix (*) that is currently used.

Reply: Asterix (*) was replaced with alphabetical letter in all tables.

Discussion

Put the subheading for the discussion section like results. Also, please keep a sequence in subheading for investigated factors, in material and method, result, and discussion.

Reply: investigated factors in the subheading were kept a sequence in all section according to your request. However, considering the coherence among paragraphs, I think it would be better to remove the subheading from the discussion section.

As a general comment: the size of fish and the fish meal content in the diets play a crucial role in fish processing. When you compare your results with other studies, to avoid any misconception, please add the fish meal contents of diets in their studies.

Reply: fish meal contents have been added in some studies, however, other studies didn’t point out the ingredients of experimental diet.

Some parts of the discussion are better updated with research in 2021 and 2020 as they refer to some old references. Please update the discussion with the latest studies as much as possible.

Reply: some references ([25], [27], [49]) have been updated.

Please clarify which fish species throughout the discussion when you say “fish”.

Reply: the academic editor pointed out that the fish species should be deleted in his revision, his original words were “particular fish species names are not essential”.

Line 321-322 to the best of our knowledge, no prior study has reported the effect of feed processing technique on intestinal microbiota for any aquatic animal. We found no significant difference in diversity (reflected by the Shannon and Simpson indexes) of crayfish.

Reply: this part has been revised according to your suggestion.

Comment: Include Table number or Figure number where that data is presented

Reply: table number and figure number have been added in the discussion section.

Conclusion

Finally, the article must address the wider relevance of your results, which must be easily understood and summarized in conclusions. Conclude based on the level of the treatment.

Reply: the conclusion section has been revised.

 Once again, thank you very much for your comments and suggestions.

Round 2

Reviewer 2 Report

all major comments have been addressed .

Author Response

Once again, thank you very much for your comments and suggestions.